# In Drosophila Hemolymph, Serine Proteases Are the Major Gelatinases and Caseinases

**DOI:** 10.3390/insects15040234

**Published:** 2024-03-28

**Authors:** Jean-Luc Gatti, Séverine Lemauf, Maya Belghazi, Laury Arthaud, Marylène Poirié

**Affiliations:** 1Université Côte d’Azur, INRAE, CNRS, Institut Sophia Agrobiotech, 06903 Sophia Antipolis, France; severine.lemauf@inrae.fr (S.L.); laury.arthaud@inrae.fr (L.A.); marylene.poirie@inrae.fr (M.P.); 2Marseille-Protéomique (MaP), Plateforme Protéomique, Institut de Microbiologie de la Méditerranée UMR 3479 CNRS, Aix-Marseille Université, 13402 Marseille, France; mbelghazi@imm.cnrs.fr

**Keywords:** Drosophila, hemolymph, zymography, proteases, gelatinase, caseinase, immunity

## Abstract

**Simple Summary:**

Insect hemolymph transports, exchanges, and eliminates soluble compounds from the hemocoel. These processes play a critical role in many physiological processes, including development and immunity. Since matrix metalloproteases (MMPs) are the major circulating gelatinases in the blood of various species, including mammals, we used gel zymography to analyze gelatinase and caseinase activities in Drosophila larval hemolymph under normal and pathological conditions. Our investigations demonstrate that the major gelatinases and caseinases in Drosophila larval hemolymph are not MMPs but serine proteases (SPs). We identified more than 60 SPs in these proteolytic active bands. While a role in immunity has been suggested for some of these circulating SPs, the physiological functions of most of them remain to be elucidated.

**Abstract:**

After separation on gel zymography, *Drosophila melanogaster* hemolymph displays gelatinase and caseinase bands of varying sizes, ranging from over 140 to 25 kDa. Qualitative and quantitative variations in these bands were observed during larval development and between different *D. melanogaster* strains and Drosophila species. The activities of these Drosophila hemolymph gelatinase and caseinase were strongly inhibited by serine protease inhibitors, but not by EDTA. Mass spectrometry identified over 60 serine proteases (SPs) in gel bands corresponding to the major *D. melanogaster* gelatinases and caseinases, but no matrix metalloproteinases (MMPs) were found. The most abundant proteases were tequila and members of the Jonah and trypsin families. However, the gelatinase bands did not show any change in the tequila null mutant. Additionally, no clear changes could be observed in *D. melanogaster* gel bands 24 h after injection of bacterial lipopolysaccharides (LPS) or after oviposition by *Leptopilina boulardi* endoparasitoid wasps. It can be concluded that the primary gelatinases and caseinases in Drosophila larval hemolymph are serine proteases (SPs) rather than matrix metalloproteinases (MMPs). Furthermore, the gelatinase pattern remains relatively stable even after short-term exposure to pathogenic challenges.

## 1. Introduction

Insect hemolymph circulating in the hemocoel is in direct contact with the tissues and organs, playing the role of both blood and lymph in mammals. Hemolymph is responsible for the circulation, transport and excretion of various classes of compounds (from gases and ions to macromolecules) and thus plays a crucial role in maintaining the homeostasis of the organism [1,2]. Hemolymph is also involved in many hormonal processes, intercellular signaling, interorgan communication, and plays a key role in immune defense through circulating humoral immune factors and hemocytes [3,4]. For example, recognition of aggressor molecules by hemolymph-specialized receptors triggers various immune signaling pathways leading to the expression of cytokines, chemokines, and antibacterial molecules [3,5,6,7]. This also involves the production of reactive oxygen and nitrogen species, such as those derived from the melanization process resulting from the activation of the pro-phenoloxidase cascade [4,8,9]. Hemocytes are also essential for innate immunity, they participate in the physical elimination of apoptotic cell remnants and of bacteria and fungi, and are involved in bacterial nodulation and in the encapsulation process of large invaders such as nematodes and parasitoid eggs or larvae [4,8,9,10].

In mammalian fluids (blood, cerebrospinal fluid, seminal plasma, etc.), the major circulating proteases are metalloproteases, in particular matrix metalloproteases (MMPs; metzincins), which are involved in the proteolytic cleavage of extracellular matrix proteins such as fibronectin and collagen [11]. MMPs have been studied for their role in numerous human pathologies such as tumor formation and metastasis, and more than 20 MMPs have been described in vertebrates [11,12]. Two of them, MMP-2 and MMP-9, are also known as gelatinase A and gelatinase B, respectively, because they degrade denatured collagens or gelatins [13,14,15]. These pro-enzymes are activated by other proteases, such as trypsin and other MMPs, and by reactive oxygen species (ROS). Metalloproteases with gelatinase activity have also been described in the blood of rainbow trout [16] and in the whole hemolymph of the mussel [17], suggesting a possible evolutionary conservation.

In *Drosophila melanogaster*, there are only two MMP genes, *DmMmp1* and *DmMmp2*, which produce proteins that can be bound to the cell membrane by a GPI anchor or can be free in the environment [18,19]. Both MMPs are required for viability, participate in the remodeling of postembryonic tissues, and have been implicated in many other physiological functions such as wound repair and the accumulation of plasmatocytes at the site of injury [20]. In vitro, these MMPs have different substrate affinities; for example, DmMmp2 cleaves gelatin but not DmMmp1, and conversely, DmMmp1 cleaves casein but not DmMmp2 [19,21]. To our knowledge, only one publication on *D. melanogaster* has shown a gelatin zymogram of a whole extract from a wild-type third instar larva. The authors suggested an increase in secretion of a 49 kDa gelatinase immunologically related to human gelatinase A (MMP2) induced by the giant lethal larva gene mutation (l(2)gl) [22]. Several gelatinases around 25–30 kDa were also observed, but no attempt was made to identify them. In the gray flesh fly *Neobellieria bullata*, zymography allowed the detection of hemolymph gelatinolytic bands with different molecular weights (ranging from 20 to >200 kDa) and showed that specific changes in the bands occurred in relation to sex and developmental stages [23]. Partial inhibition by EDTA suggested that some of these *N. bullata* gelatinolytic enzymes are metalloproteases.

Thus, since no study has been conducted to analyze the gelatinolytic and caseinolytic activities of Drosophila hemolymph, here we used gel zymography, the most widely used and sensitive tool to reveal the presence, activity, and isoforms of gelatinases [15], biochemistry and mass spectrometry to study the gelatinases and caseinases in Drosophila larval hemolymph.

## 2. Materials and Methods

### 2.1. Chemicals

Acrylamide, agar, porcine gelatin, milk casein, protease inhibitors, EDTA, and all other reagents were of the best quality available (Sigma, Saint-Quentin-Fallavier, and Carl Roth, Lauterbourg), France). Zymographic gels were stained with Coomassie brilliant blue (CBB) (PhastGel^®^ blue R; GE Healthcare, Uppsala, Sweden) according to the manufacturer’s instructions. Molecular weight standards for electrophoresis ranged from 10 to 250 kDa (PageRuler™ Plus, Thermo Fisher, Bourgoin-Jallieu, France).

### 2.2. Biological Material

The *Drosophila melanogaster* strains used were as follows: the wild-type “Nasrallah” from Tunisia (Gif stock 1333, Gif-sur-Yvette, France), “Brazza” from the Republic of Congo (Gif stock 1588), a French field-collected strain “Sefra” collected in southern France, the YR strain (Gif stock 1088), and the YS strain (Gif stock 1089) [24]. *Drosophila melanogaster* Canton S, the yellow and white double mutant YW118, and the iso-1 strain (tequila deleted mutant; stock #2057) [25] were obtained from the Bloomington Drosophila Stock Center (Bloomington, IN, USA). Other Drosophila species included *D. simulans* (Gif stock 1132), *D. yakuba* (Gif stock 1880), *D. immigrans* (a French field strain), and the invasive *D. suzukii* [26]. All flies were maintained on standard medium (10% organic cornmeal, 10% yeast, agar and nipagin) at 25 °C with a 12 h light/dark cycle.

Two strains of the endoparasitoid *Leptopilina boulardi* (Hymenoptera, Figitidae) were used for parasitism and venom experiments: ISm (Gif stock G431) and ISy (Gif stock G486) [24]. The parasitoids were reared at 25 °C on the *D. melanogaster* Nasrallah strain susceptible to these parasitoids. After emergence, adults were maintained at 20 °C on agar medium supplemented with honey. All experiments were performed with 5–10-day old mated females. Two females were allowed to parasitize 20–40 young L2 larvae for 2 h. The larvae were kept 24 h at 25 °C and half of the larvae were used for hemolymph collection, while the other half was dissected 48 h later to measure the parasitism rate. At least two thirds of the larvae were parasitized during the experiments.

To test the direct effect of ISm and ISy venom on the hemolymph, female wasp venom reservoirs were obtained by dissection and diluted in Insect Ringer (IR) solution (KCl, 182 mM; NaCl, 46 mM; CaCl2, 3 mM; Tris HCl, 10 mM; pH 7.2) as previously described [27]. After centrifugation at 500× *g* to remove tissue debris, 20 µL of the supernatant (containing the extract from 5 reservoirs) was mixed with 10 µL containing the hemolymph of 10 L2 larvae. After 20 min at RT, samples were mixed with non-reducing sample buffer prior to gel zymography.

Solution of LPS (5 mg/mL in IR) from *Escherichia coli* 055:B5 and 0111:B4 (L2880 and L3024, respectively; Sigma, Saint-Quentin-Fallavier, France), *Salmonella enterica* serotype enteritidis (L6011; Sigma), and *Serratia marcescens* (L6136; Sigma) were separately microinjected (60 nl; FemtoJet; Eppendorf, Montesson, France) into L2 larvae (>20). Hemolymph was collected 24 h after injection. Hemolymph from larvae injected with IR was used as a control.

### 2.3. Larva Tissue Collection

For hemolymph collection from L2 larvae (3 days old), young and late L3 larvae (4 and 5 days old, respectively) were washed in 70% ethanol and rinsed twice in IR. The dorsal anterior cuticle was gently torn with forceps in a drop of IR. Larvae needed for one experiment were successively exsanguinated in the same drop. The hemolymph solution was centrifuged at 500× *g* for 10 min to remove hemocytes and debris, and the supernatant, considered cell-free hemolymph, was processed immediately. The number of larvae used is indicated in the legend of each figure.

Complete intestinal tracts from 10 L2 larvae starved for 30 min were dissected and stored in 20 µL IR before being thoroughly crushed with a pestle in a microtube. After centrifugation at 10,000× *g* for 10 min, the supernatant (~15 µL) was mixed with non-reducing sample buffer.

### 2.4. In Gel Zymography

We used 12.5% SDS polyacrylamide gels for protein separation. For zymography, the separating gel solution was supplemented with 0.15 mg/mL porcine gelatin or milk casein. Additionally, 2D gel protein isoelectrofocalization and zymography were performed as previously described [28]. After migration, zymograms were washed for 1 h in a water solution containing 2.5% (*w*/*v*) Triton X-100 to allow in situ protein renaturation and incubated for 18 h at 30 °C in PBS. Enzymatic activity appeared as white bands or spots on a blue background after Coomassie Brilliant Blue (CBB) staining. For visualization of protein bands or spots, equivalent gels without gelatin were silver stained [29].

For the inhibitor assay, the hemolymph sample was evenly loaded on a 12.5% gelatin gel fitted with one tooth comb stacking gel. After running, the gel was cut into ~1 cm wide slices and each slice was washed and incubated as described above with one of the following inhibitors at the indicated concentrations: EDTA, 1 mM; E64 (trans-epoxysuccinyl-L-amido(4-guanido)butane), 5 μg/mL; AEBSF (4-(2-aminoethyl)benzenesulfonyl fluoride), 1 mM; and leupeptin, 0.5 μg/mL. After 18 h at 37 °C, the zymogram section was stained with CBB, and inhibition was visualized by the absence or decrease in intensity of the protease band compared to a control lane incubated in PBS. Gels and stained sections were photographed (Canon markII; Canon, Tokyo, Japan), and the color digital images were cropped for presentation and processed with image processing software (Graphic converter v12; Lemke Software, GmbH, Peine, Germany) to correct contrast for better visualization of bands on zymography.

Gelatinase band intensity analyses were performed using ImageJ software (https://imagej.nih.gov/ij; V2.14.0/1,54f; 7 July 2023). Briefly, the unprocessed .jpg image was converted to 16-bit gray, and the intensity of the band was estimated by the total pixel intensity of a rectangular box drawn around the band and subtracted from the average background intensity measured using the same box area near the band. The intensity was divided by the pixel area of the box and expressed in arbitrary units (AU).

### 2.5. Gelatinase Purification

Hemolymph from 50 late L2 *D. melanogaster* Nasrallah larvae were pooled in 120 µL of IR, centrifuged 10 min at 500× *g* and 100 µL of supernatant was collected, diluted 1/5th with 400 µL of Tris 20 mM pH 7.5 before concentration back to 100 µL on a 10k cut-off centrifugal filter (Amicon, Merck Chimie, France). Next, 20 µL of the reconcentrated hemolymph were removed to be used as control. The remaining desalted hemolymph was then mixed with ~500 µL of DEAE-Sepharose, which was previously thoroughly washed in Tris-HCl 20 mM pH 7.5. After 10 min under mild agitation, the beads were centrifuged (500× *g*, 30 s) and the supernatant stored on ice. The beads were then washed three times in 20 mM Tris-HCl pH 7.5 and resuspended in 400 µL of 50 mM NaCl/Tris-HCl 20 mM pH 7.5 for 10 min under mild agitation. The beads were centrifuged, and the supernatant was removed and stored on ice. This step was repeated with increasing NaCl concentration (250, 500 and 1000 mM in Tris-HCl 20 mM). Each eluate was further concentrated down to 40 µL using 10 kDa cut-off centrifugal filter devices before mixing with non-reducing sample buffer (20 µL). The same protocol was applied with gelatin-sephadex beads (Cytiva, Saint-Germain-en-Laye, France). Half of each concentrated elution volume was loaded onto a 12.5% SDS-PAGE gel, and the other half was loaded onto a gelatin-containing 12.5% SDS-PAGE. Alternatively, for gelatin beads after hemolymph binding and washing steps, instead of salt desorption, the beads were directly treated with 50 µL of non-reducing sample buffer to ensure that all proteins were extracted. The same binding protocol was used with benzamidine Sepharose 4FF beads (Cytiva, Saint-Germain-en-Laye, France), but after binding and washing, the elution was performed with 20 mM of 4-benzamidine in Tris 20 mM, pH 7.5 and then the beads were mixed with non-reducing sample buffer to extract the still bound proteins. All experiments were performed at least twice.

### 2.6. Mass Spectrometry

Silver-stained gels and zymograms were used for mass spectrometry sequencing. The center of the protease activity area and/or the corresponding band or spots from the normal gel were carefully excised and stored at −20 °C prior to analysis.

Protein identification was performed by LC-MS/MS (Q-orbitrap mass spectrometer, Q-ExactiveTM Plus, Thermo-Fisher Scientific, Bourgoin-Jallieu, France). Spots and excised bands were treated with porcine trypsin (12.5 ng/μL; Promega, Charbonnières-les-Bains, France) to generate protein fragments. Peak lists were generated with Proteome Discoverer 3.0 in .mgf format and analyzed with in house Mascot 2.3 software (Matrix Science Ltd., London, UK) using Uniprot *D. melanogaster* (downloaded March 2021) (Appendix A). Mascot Mudpit scoring parameters were used (sum of the scores above the threshold of significant peptide matches plus the average threshold of these matches). The significance threshold was set to *p* < 0.05, and the maximum number of hits was set to AUTO (to display all hits with a protein score above the average identity threshold score for a single peptide match). An automated search of the decoy database was performed. The analysis was performed with a mass tolerance of 20 mDa for fragment ions and 10 ppm for parent ions. Carbamidomethylation of cysteines and oxidation of methionine were used as variable modifications for the calculation of peptide masses. The maximum number of trypsin-missed cuts allowed was 2. Data were also searched using in house Proteome Discoverer 3.0 and the Chimerys node (Thermo-Fisher Scientific, Bourgoin-Jallieu, France) (Appendix A). Default parameters were used to search the *D. melanogaster* database from Uniprot (www.uniprot.org/, 7 July 2023) downloaded in June 2023. Protein hits were annotated using the default consensus workflow CWF_Comprehensive_Enhanced_Annotation using the same database and then filtered using the GO accession number GO:0008236 corresponding to the molecular function serine-type peptidase activity. Only high-confidence proteins that met the high FDR threshold of 1% and were identified with at least two peptides were conserved. All MSMS data used can be found in Appendix A. The raw proteomics data are deposited in the PRIDE archive (www.ebi.ac.uk/pride/archive/, 7 July 2023). The project name is: *Drosophila melanogaster* hemolymph gelatinase; project accession number: PXD045527.

### 2.7. Tequila Expression

Total RNA was obtained from 30 larvae crushed in 350 µL lysis buffer according to the manufacturer’s instructions (RNeasy Plus Micro Kit, Qiagen, Les Ulis, France). Total RNA was then treated with 1 unit/µg DNase (RQ1 RNase-Free DNase; Promega, Charbonnières-les-Bains, France) for 10 min at 65 °C to remove any residual gDNA. cDNA synthesis was performed using a Superscript IV reverse transcriptase kit (Invitrogen, Illkirch, France). For PCR, cDNA was mixed with nucleotides and GoTaq G2 DNA polymerase (Promega, Charbonnières-les-Bains, France) and 20 µM Tequila forward primer AGTGACTATGTGCAGCCCAT and reverse primer TTCACAGGCATCCACACTCT and as PCR control RP49 (ribosomal protein L32) forward primer CGCACCAAGCACTTCATC and reverse primer CACTCTGTTGTCGATACCCTTG (denaturation, 94 °C, 3 min; annealing and extension, 30 cycles, 94 °C, 30 s and 58 °C, 45 s; final extension, 72 °C, 8.5 min).

### 2.8. Statistical Analysis

Data presentation and analysis were performed using PRISM 10 software (GraphPad Software, Boston, MA, USA). We used two-way ANOVA followed by Tukey’s multiple comparison tests for data obtained from each measured band under different conditions. Three to five biological replicates were used in all experiments performed in this study to ensure representativeness and statistical power.

## 3. Results

### 3.1. Gelatinases in *Drosophila melanogaster* Larval Hemolymph

Hemolymph collected from the wild type *D. melanogaster* Nasrallah larvae at different stages (L2, young L3 (yL3) and wandering (L3)) was separated on a one-dimensional zymogram under non-denaturing conditions using gelatin as substrate (Figure 1a). Several active gelatinolytic bands (white bands on the blue background) were observed, forming a smear at the top of the gel. At least six distinct bands (labeled a to f) of varying intensity ranging from 60 to 25 kDa were observed. The smear at the top of the gel indicates the precipitation of some proteins or a high-molecular-weight form or complex that did not enter the resolving gel. In the late L3 hemolymph, a large dark blue-stained protein of approximately 70 kDa represents the Larval Serum Proteins (LSPs; 78–83 kDa), highly secreted into the hemolymph at this larval stage [30,31], that may interfere with the visualization of the 60- and 55-kDa gelatinase bands. The caseinase activity of *D. melanogaster* Nasrallah L2 larval hemolymph was also tested (Figure 1b). It was lower compared to the gelatinase activity, and the main caseinolytic bands were observed at approximately ~38 kDa (C3; equivalent to gelatinase d) and ~40 kDa (equivalent to gelatinase c). Additionally, a fainter band at ~60–65 kDa (equivalent to gelatinase a) and large smears at the top of the gel were observed (C1-C2).

To verify that the gelatinases were specific to hemolymph, we first compare gelatinases in this fluid and that of the whole larval extract and found that the band pattern was clearly different (Appendix A). Additionally, when intact larvae were washed and either left in Insect Ringer solution (IR) for 10 min or centrifuged in IR for 5 min at 5000× *g*, the solution showed only a gelatinolytic band of approximately 80 kDa and a faint band of approximately 50 kDa (Appendix A), indicating minimal contamination from larval excretion. Finally, we compared the gelatinase profiles of the hemolymph and gut extract that is rich in proteases [32]. The gelatinase profile of the gut extract differed from that of the hemolymph (Appendix A).

The representativeness of the wild-type Nasrallah strain of *D. melanogaster* was tested by comparison with six additional strains of this fly species (Figure 1c). Although the general pattern of hemolymph gelatinases looks more or less similar among strains, the relative intensity of the main gelatinase bands varied strongly from strain to strain. While we observed variations between gels, particularly with the sharp increase in bands (e) and (f) (see Appendix A), it seems that the Sefra and Canton S lines had lower overall gelatinase activities.

### 3.2. Larval Hemolymph Gelatinases in Different Drosophila Species

Stage 2/3 larvae of *D. melanogaster* (Canton S and Nasrallah strains), *D. simulans*, *D. yakuba*, *D. immigrans*, and the invasive *D. suzukii* were used to compare hemolymph gelatinases. *D. suzukii* showed three main bands at ~40, ~100, and ~120 kDa. *D. yakuba*, one strong band was visible at ~90 kDa, and fuzzy activity areas were observed ~200 and ~60 kDa (Figure 2; see also Appendix A). The hemolymph of *D. simulans* showed a single low-intensity band at 50 kDa, while that of *D. immigrans* had very low activity with only one faint band at about 80 kDa. This indicates that Drosophila species hemolymph contain different proteases, as evidenced by their distinct gelatinase patterns.

We noticed that some of the gelatinolytic bands in *D. melanogaster* (mainly band a and b) were less affected than others by the presence of the reducing agent β-mercaptoethanol in the sample buffer (Appendix A). However, none of them resisted heat denaturation. We also observed an unusual behavior of *D. melanogaster* hemolymph proteins; as the concentration of the reducing agent in the sample buffer increased, there was a decrease in high-molecular-weight proteins (Appendix A). This phenomenon was not observed in other Drosophila species.

### 3.3. D. melanogaster Hemolymph Gelatinase and Caseinase Activities Are Inhibited by Serine Protease Inhibitors

To determine the class of hemolymph gelatinases, we tested different types of protease inhibitors: E64 for cysteine proteases (including cathepsins), AEBSF for serine proteases, leupeptin for both serine and cysteine proteases, and EDTA for metalloproteases. For *D. melanogaster* Nasrallah and Canton S, a large decrease in intensity occurred with AEBSF or leupeptin, but no effect was observed with EDTA and E64 (Figure 3a; Table 1). We estimated the change in the intensity of some of the gelatinase bands of the *D. melanogaster* Nasrallah strain (Figure 3b). For the four gelatinolytic bands (a, b, d, f), the decrease in intensity in the presence of AEBSF and leupeptin was statistically significant compared to the control (Appendix A). Similar to gelatinase, the caseinase bands from *D. melanogaster* hemolymph were also affected by AEBSF (Appendix A).

This inhibitor panel was used on the hemolymph gelatinase of different Drosophila species, and a visual estimation of inhibition on all bands indicated that, for all Drosophila species, the main hemolymph gelatinolytic activities were due to serine proteases (Table 1).

### 3.4. Identification of D. melanogaster Hemolymph Putative Gelatinases and Caseinases

First, we directly cut the hemolymph gelatinase and caseinase bands from the zymography gels for mass spectrometry (Appendix A). As gelatinases and caseinases in hemolymph were inhibited by serine protease inhibitors, we focused on this type of protease present in each band even with a low mascot score threshold (>50). Unfortunately, certainly due to the gelatin presence, only a few protein identifications per band could be obtained, and none corresponded to a protease. Therefore, we cut the equivalent bands to the main gelatinase and caseinase bands on the corresponding non-reducing silver-stained gels (Figure 4) and analyzed their content by mass spectrometry (Appendix A).

In the four gelatinase-equivalent bands (G1–G4 corresponding to gelatinase bands a, b, c and f, respectively) and the three caseinase-equivalent bands (C1–C3; notice that band C3 may correspond to gelatinase band d), a large number of known hemolymph proteins were identified by mass spectrometry (Appendix A) in agreement with previous proteomic studies [33,34,35]. In the bands G1–G4, a total of 31 different serine proteases (trypsin, chymotrypsin and elastase types) were found, and the most represented families were the Jonah proteases (nine members) and trypsin (four members) (Table 2).

Some of the gelatinase bands from the *D. melanogaster* Canton S line hemolymph were also analyzed and gave a similar protease panel (Appendix A, Appendix A). A 2D gel was also performed using the *D. melanogaster* Nasrallah hemolymph. The equivalent position of the strongest gelatinase spots at ~55 kDa on the silver-stained gel allowed for the identification of only the tequila serine protease (Appendix A).

For the three caseinase equivalent bands C1–C3, 19 serine proteases were retrieved and nine were from the Jonah family (Table 3). Jon65Aiv was present in the three bands, whereas tequila and SP21 were present in two bands. The 16 other proteases were all in band C3.

We attempted to enrich hemolymph gelatinases using chromatography. On DEAE-sephadex, the main gelatinase band eluted was at 25 kDa (Appendix A) and could be the equivalent of band f. Mass spectrometry analysis identified only the elastase-like SP151 (CG18180), which is also the highest score for this protease found only in band G4/f. After purification on gelatin beads, only a few fuzzy gelatinase activities were observed after salt elution, and after silver staining, the only visible bands that were purified were at ~35 and ~70 kDa, with no apparent activity (Appendix A). Only when the beads were treated with non-denaturing sample buffer a ~65 kDa gelatinase band was observed, but without a counterpart on the silver-stained gel. We then attempted to purify the serine proteases using benzamidine-Sepharose beads. After incubation and washing, the beads were eluted with 20 mM para-aminobenzamidine (PABA) to release the bound proteases. While many proteins appeared specifically concentrated in the eluate, almost no gelatinase activity was observed (Appendix A). When PABA-treated beads were further washed in non-reducing sample buffer to remove all bound proteins, the profile of these proteins was close to that of PABA-eluted proteins, and gelatinase activities could be observed at ~35, ~40 and ~65 kDa (Appendix A). Because the lack of activity in the PABA-eluted proteins was certainly due to an inhibition effect of PABA, the eluted bands and the bands released by the sample buffer were cut from the silver-stained gel and subjected to MS-MS analysis (Appendix A; Appendix A). Among the PABA-eluted proteins, various SPs such as the epsilon and beta isoforms of Trypsin, Tequila and MP1 were identified. From gel bands of beads treated with sample buffer, the serine proteases Jon99Cii (SP1), Tequila, SP7, MP1 (SP25), Trypsin zeta (SP110), SP34 (CG9372), SP83 (CG17571), SP151 (CG18180), Jon66Ci, SP78, Trypsin epsilon, and yip7 were identified (Appendix A).

To further explore the MS data from the gelatin and casein gel bands and the equivalent bands G1–G4 and C1–C3, we used the AI-powered Chimerys software (Appendix A). This allowed us to identify more than 7500 proteins with a high confidence score and found more than 60 serine-type peptidases. Almost all the serine proteases found with the Mascot analysis were retrieved using Chimerys, and new ones were found, such as SP22 or Easter. Tequila and Jon65Aiv were also the two most abundant proteases in the Chimerys analysis (Appendix A).

### 3.5. Hemolymph Gelatinase Activities in D. melanogaster Tequila KO Larvae

Since *tequila* was among the major proteases observed in our samples, we tested the hemolymph gelatinase of the *tequila* null mutant iso-1 line (BDSC #2057) (Appendix A). The overall pattern of L2 hemolymph gelatinases did not appear to be affected compared to wild-type (Appendix A).

### 3.6. LPS Injection Did Not Affect Gelatinase Activity

Since many humoral proteases are involved in the immune response, we tested the effect of injection of bacterial lipopolysaccharide (LPS) known to produce an immune reaction [36,37]. *D. melanogaster* Nasrallah L2 larvae were injected with different LPS and the hemolymph was collected 24 h later (stage yL3). Although we observed up to 50% mortality of larvae 24 h after LPS injection, we did not observe a clear trend in the gelatinase activities in the hemolymph of surviving larvae. The estimated intensity of gelatinase bands a and d from three different experiments showed very large variations and no significant differences or trends from the controls (Figure 5; Appendix A).

### 3.7. Apparent Lack of Wasp Parasitism Effect on Gelatinases

When a female *Leptopilina boulardi* of the Ism or Isy strains parasitizes a *D. melanogaster* Nasrallah larva, the parasitoid success rate is always high. However, in the *D. melanogaster* YR line, parasitism by the Isy strain fails mainly due to encapsulation of the parasitoid eggs [24,38,39]. When the hemolymph of *D. melanogaster* Nasrallah or YR larvae was collected 24 h after ISm or ISy parasitism, no clear reproducible changes in gelatinases occurred compared to the control condition (Appendix A), suggesting that there was no visible effect of the parasitoid larva or the encapsulation process on gelatinase activities during this time period. Pre-incubation of *D. melanogaster* Nasrallah hemolymph with ISm or ISy parasitoid venom also had no visible effect on gelatinase bands (Appendix A).

## 4. Discussion

The hemolymph of Drosophila larvae contains several bands with gelatinase and caseinase activities. The hemolymph pattern of gelatinolytic bands was different from those of the larval extract or released from larval secretion/excretion, suggesting that they were not the result of contamination. The presence in the mass spectrometry analysis of known Drosophila hemolymph proteins, such as LSPs, PPO1 and PPO2, as well as extracellular SPs [9,40,41,42], is also a good indication that we collected a rather clean hemolymph. We noticed that quantitative and qualitative differences exist between *D. melanogaster* strains and Drosophila species. While differences between species could be expected, these between *D. melanogaster* lines were more surprising. However, studies on *D. melanogaster* populations indicated adaptive changes in gene expression depending upon the geographic and even the laboratory origin, and among these genes many were involved in proteolysis, including trypsin and Jonah family members [43]. It is also important to note that the different strains and species have different larval developmental time that can influence the gelatinases expression pattern.

We were also surprised by the variations in the intensity of the *D. melanogaster* gelatinase bands between experiments with, in many cases, a decrease in the intensity of the high MW gelatinase bands and an increase in the low MW bands. Because we diluted the hemolymph during collection and removed cells by centrifugation, we expected that if a clot formed rapidly and trapped proteins (and proteases) [34,44], it would be removed, but this removal process takes time and may allow for some proteolysis and autoactivation. This proteolysis is also suggested by the fact that the same protein (including proteases) can be found in the different gel bands analyzed by MS-MS. Thus, although degradation or conversion of high-MW compounds may occur, since we treated all samples in the same way in an experiment, we expected them to have virtually the same degree of degradation/conversion. We also observed a striking effect when we treated *D. melanogaster* hemolymph proteins with a reducing agent. The reason for this remains to be determined, but could indicate a higher degree of proteolysis of the circulating proteins after collection in this species (may be related to more proteases), or alternatively the formation of covalent inter-protein bonds that render them insoluble, although these two effects are not exclusive. These experimental variations and the fact that changes in hemolymph proteases and protein composition occur during the development of *D. melanogaster* larva made long-term analysis difficult, as was the case with the effects of LPS or parasitism, which were measured 24 h after treatment.

Overall, we identified more than 60 SPs and ten serine protease homologs (SPHs); some of the SPs are known members of the extracellular immune signaling network, while many others are predicted to be secreted but have unknown functions. This high number is consistent with recent publications analyzing *Drosophila* SPs including those in the adult hemolymph [45,46]. Among the *D. melanogaster* serine proteases identified here, the proteases tequila, and Jonah and trypsin members were the most redundant. Tequila is described as a neurotrypsin involved in long-term memory and insulin signaling in adult flies [47], however, in the larva, it is mainly expressed in the fat body (FlyAtlas2). This multi-domain serine protease is a large protein of 260 kDa with a C-terminal serine protease domain [25]. The null mutant shows no phenotype, but in the wild-type fly, up-regulation of *tequila* expression is observed 24 h after fungal or bacterial infection [25] and in the embryo after wounding [48], suggesting a role in tissue remodeling. The tequila protein was also previously found by proteomics in fly hemolymph and no significant changes were observed after bacterial, viral or fungal infection [49]. No major change in the larval hemolymph protease bands was observed with the *tequila* null mutant, suggesting that although this protein is quantitatively present, none of the major bands may be an active fragment. This may also explain why we did not observe a change in gelatinases pattern linked to tequila variation after the immune challenges, although this needs a further exploration.

The Jonah gene family consists of 19 genes distributed in small clusters at different chromosomal locations [50]. These ~30 kDa proteases from different classes are involved in many different physiological functions with distinct spatial and temporal expression during Drosophila development; these genes are expressed at all larval stages, disappear at the end of L3, and some reappear in the adult midgut, fat body, ring gland, imaginal discs, and salivary glands [50,51]. Consequently, their presence in these different tissues suggests that the Jonah proteases may be secreted in the hemolymph [50]. Jon65Aiv has been shown to degrade unfolded proteins and in vivo may prevent the intracellular accumulation of polyubiquitylated proteins in larvae [52]. Jon66Ci regulates immune signaling in the gut of Drosophila larvae, but also in other immune tissues such as the fat body or hemolymph [53].

Trypsins are known to exhibit gelatinolytic activity [54]. While their predictive MW is around 25–30 kDa, here they were found in higher MW gelatinase and caseinase bands, suggesting that they may form multimer or complex with other proteins. Alternatively, their migration in the gel may be retarded by their affinity/binding to gelatin. Only the hemolymph gelatinase band f could match the predicted MW, and indeed three members of the trypsin family were found in the corresponding band. Trypsin genes, like Jonah genes, are organized in tight genomic clusters and are expressed sequentially along the intestine [55]. However, perivitelline injection of trypsin induced a global epidermal wound response, possibly by processing and activating a ligand that initiates a wound response pathway [48]. An increase in trypsin activity after mating has been also demonstrated in female *D. melanogaster* hemolymph [56].

Other serine proteases present in our analysis may also have gelatinase/caseinase activity, most of them are highly expressed during the larval stages but physiological data are only available for a few of them. One example is SP34 (CG9372), this trypsin-like protease is reported to be expressed in the larval trachea [57] and ring gland [52] and to be involved in response to infection [58].

In our assays, we did not observe significant changes in gelatinase or caseinase activities after immune challenge with either bacterial LPS or after parasitism. One of the reasons may be that the zymographic assays reveal all gelatinases/caseinases already present in the hemolymph, even if they are normally under inactivated pro-form [15], then the change of SPs from pro-form to activated form may not be easy to detect particularly if a small change occurred. We also did not observe a direct effect of *Leptopilina boulardi* venom on gelatinase, suggesting either that the amount of serpin, an irreversible inhibitor, present in it is not sufficient to inhibit all gelatinolytic SPs or, more likely, that the venom serpin is highly specific for one type or even one of the Drosophila SPs [59]. Unfortunately, probably due to the peculiar behavior of *D. melanogaster* hemolymph after collection, we were not able to purify the different main gelatinases to test further this hypothesis.

Keeping in mind that zymography does not represent true in vivo biological activity of the proteases but is an overview of the proteases present, one question that arises is why are there so many gelatinases/proteases in Drosophila larval hemolymph? Gelatinase activity has been studied mainly in mammalian physiological fluids including blood, urine and reproductive fluids, and the main gelatinases found were MMPs [14,28,60]. To our knowledge, much less is still known in other taxa and even lesser in insects. Insect hemolymph has many diverse functions that may change during development. While it has been mainly studied for defense mechanisms, the main role of hemolymph is to maintain homeostasis by delivering nutrients and hormones to cells and to remove cell waste. Serine and other proteases must thus be secreted and involved in tissues remodeling by degrading extracellular components during the larval development and may be released from live and apoptotic cells during these tissues remodeling. Since Drosophila from the first larvae to the prepupal stages is in a developmental process during which the growth is exponential, an extensive tissue transformation must be undergoing, involving certainly important intra- and extracellular proteostasis [61]. Some of the circulating SPs found are suggested to participate in the digestive process, but they may also be secreted in the hemolymph or transferred through gut epithelial cells by transcytosis along with nutrients [62]. As indicated, many are also expressed in other tissues and sometime during specific larval development stage(s) suggesting they may be involved in different physiological functions than in adult fly.

## 5. Conclusions

In conclusion, our data showed that *Drosophila* larval hemolymph contains several gelatinase and caseinase bands that are mainly inhibited by serine protease (SP) inhibitors but not by EDTA, and we identified only many SPs in the major gelatinase and caseinase gel bands but not MMPs using mass spectrometry. We also observed that *D. melanogaster* hemolymph had a different behavior after collection than that of other Drosophila species. This work paves the way for future research on the origin and physiological role played by the majority of the SPs found in the hemolymph.

## Figures and Tables

**Figure 1 insects-15-00234-f001:**
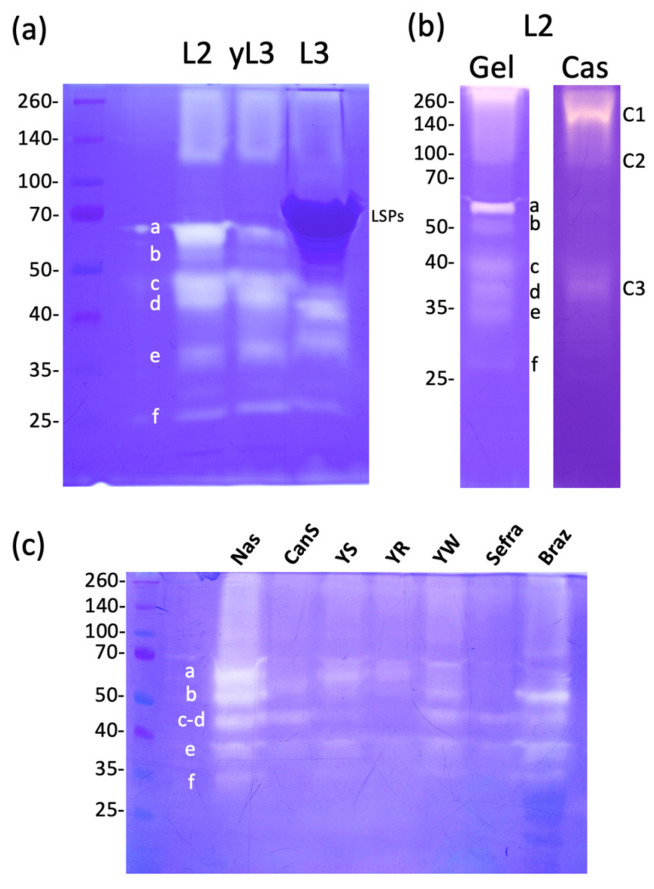
*D. melanogaster* hemolymph gelatinases. (**a**), *D. melanogaster* Nasrallah hemolymph proteins from 10 L2 larvae (L2), L3 young (yL3) and wandering larvae (L3) were separated on a 12.5% gelatin SDS-PAGE that was further incubated overnight at 30 °C in PBS and stained with CBB. The gelatinase bands (white on blue background) were labelled from a to f from their estimated molecular weight: a, ~60 kDa; b, ~55 kDa; c, ~45 kDa; d, ~40 kDa; e, ~35 kDa; f, ~25–28 kDa. (**b**), hemolymph from 20 *D. melanogaster* Nasrallah L2 larvae was split in two and separated either on a gelatin (Gel) or casein (Cas) gel; main caseinases were labeled C1, C2 and C3. (**c**), hemolymph gelatinases from 10 L2 larvae from *D. melanogaster* Nasrallah (Nas), Canton S (CanS), YS, YR, YW118 (YW), Sefra and Brazza (Braz) strains were compared. Molecular weights in kDa.

**Figure 2 insects-15-00234-f002:**
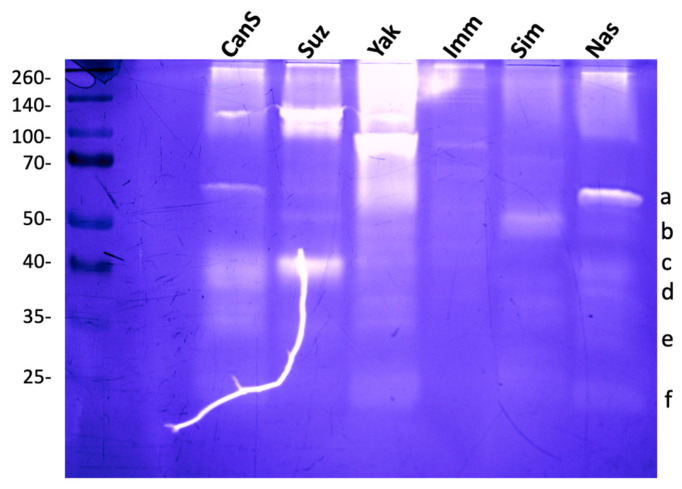
Comparison of hemolymph gelatinases from different Drosophila species. Hemolymph gelatinases from *D. melanogaster* Nasrallah (Nas) (gelatinase bands were labelled from a to f as in Figure 1b) and Canton S (CanS) strains were compared with those of *D. suzukii* (Suz), *D. yakuba* (Yak), *D. immigrans* (Imm) and *D. simulans* (Sim). Hemolymph from 10 L2 larvae was separated on 12.5% SDS-PAGE. Molecular weights in kDa. The white line is due to a crack in the gel.

**Figure 3 insects-15-00234-f003:**
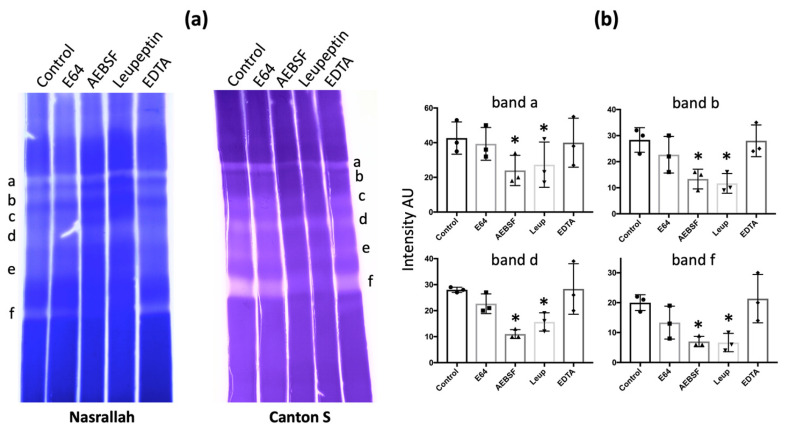
Inhibition of *D. melanogaster* gelatinases. (**a**), after migration of *D. melanogaster* Nasrallah or Canton S hemolymph proteins from 50 late L2 larvae, ~1 cm slices of the 12.5% SDS-PAGE were incubated overnight at 30 °C in PBS alone (Control) or PBS supplemented with E64, AEBSF, Leupeptin or EDTA before CBB coloration. Gelatinase bands were labelled from a to f as in Figure 1a. (**b**), estimated intensity of the gelatinase bands a, b, d, f after incubation with inhibitors: only inhibition by AEBSF and Leupeptin was statistically significant compared to the control (n = 3; * = *p* < 0.05; Appendix A). All slices shown came from the same experiment.

**Figure 4 insects-15-00234-f004:**
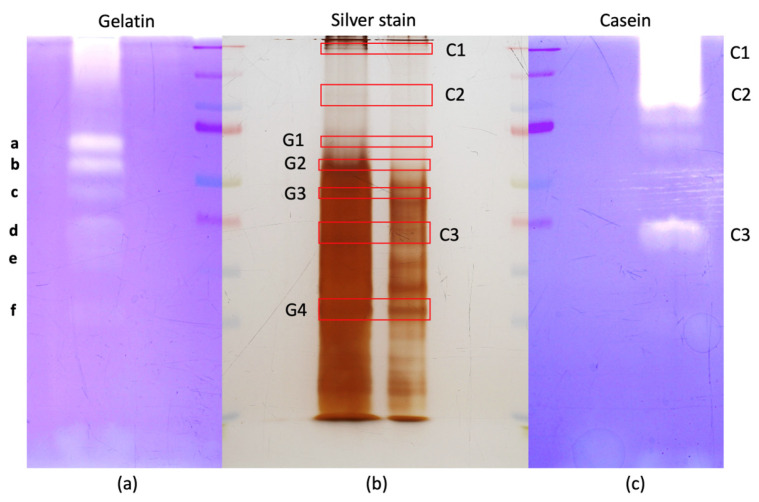
*D. melanogaster* gelatinase and caseinase bands used for MS-MS. *D. melanogaster* Nasrallah hemolymph proteins from L2 larvae (L2) were separated on a gelatin (**a**) and a casein (**c**) gel (hemolymph of 35 larvae on each) and on an equivalent gel (**b**) that was silver stained (two quantities, hemolymph of 30 and 15 larvae, respectively, were loaded on the silver-stained gel to better visualized the protein bands). Bands from the silver-stained gel that were at the same MW than the main gelatin (G1–G4) or casein (C1–C3) (red boxes) bands were cut and further processed for mass spectrometry. Gelatinase bands were labelled from a to f as in Figure 1b.

**Figure 5 insects-15-00234-f005:**
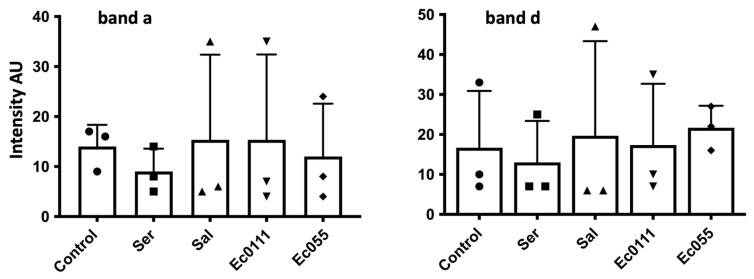
No significative change in *D. melanogaster* gelatinases after LPS injection. Intensity in arbitrary units of the hemolymph gelatinase band a (**left panel**) and band d (**right panel**) 24 h after injection of Insect Ringer (Control) or LPS from *Serratia marcescens* (Ser), *Salmonella enterica* (Sal), and *Escherichia coli* (Ec055 and Ec0111) (ten injected L2 larvae for each condition). Large variations were observed between experiments; none of the experimental treatment was significatively different from the control.

**Table 1 insects-15-00234-t001:** Inhibitors effect on Drosophila hemolymph gelatinases.

	*D. melanogaster*		*D. immigrans*	*D. yakuba*	*D. simulans*	*D. suzukii*
	Nasrallah	Canton S				
E64	−−−	−−−	−−−	−−−	−−−	−−−
AEBSF	+++	−++	+++	+++	+++	+++
Leupeptin	+++	+++	−++	−++	+++	+++
EDTA	−−−	−−−	−−−	−−−	−−−	−−−

Each symbol denotes the result of one separate experiments; −, no visible effect; +, clear visible effect.

**Table 2 insects-15-00234-t002:** Serine proteases found in silver-stained bands equivalent to *D. melanogaster* Nasrallah hemolymph gelatinase bands G1–G4 from Figure 4.

	Name	CG	Mascot Score	Type
Band G1	SP117/Jon65Aiv	CG6467	396	Chymotrypsin-like serine protease
	SP98/Jon65Aiii	CG6483	245	Elastase-like serine protease
	SP72/Tequila/Graal	CG4821	216	Trypsin-like serine protease
	SP21	CG3355	118	Trypsin-like serine protease
	SP131	CG17475	65	Chymotrypsin-like serine protease
	SP172/modSP	CG31217	63	Chymotrypsin-like serine protease
	SP119/betaTry	CG18211	56	Trypsin-like serine protease
	Hayan	CG6361	55	Trypsin-like serine protease
Band G2	SP72/Tequila/Graal	CG4821	744	Trypsin-like serine protease
	SP98/Jon65Aiii	CG6483	461	Elastase-like serine protease
	Hayan	CG6361	256	Trypsin-like serine protease
	SP117/Jon65Aiv	CG6467	211	Chymotrypsin-like serine protease
	SP172/modSP	CG31217	148	Chymotrypsin-like serine protease
	SP21	CG3355	137	Trypsin-like serine protease
	SP91	CG16749	103	Chymotrypsin-like serine protease
	SP131	CG17475	76	Chymotrypsin-like serine protease
	SP119/betaTry	CG18211	71	Trypsin-like serine protease
	SP222	CG30187	70	Trypsin-like serine protease
	SP88/epsilonTry	CG18681	69	Trypsin-like serine protease
	SP143/Jon99Ciii	CG31362	54	Chymotrypsin-like serine protease
Band G3	SP72/Tequila/Graal	CG4821	816	Trypsin-like serine protease
	SP98/Jon65Aiii	CG6483	460	Elastase-like serine protease
	SP143/Jon99Ciii	CG31362	248	Chymotrypsin-like serine protease
	SP119/betaTry	CG18211	216	Trypsin-like serine protease
	SP117/Jon65Aiv	CG6467	139	Chymotrypsin-like serine protease
	SP102/alphaTry	CG18444	114	Trypsin-like serine protease
	SP21	CG3355	104	Trypsin-like serine protease
	SP29	CG14642	83	Trypsin-like serine protease
	psh/persephone	CG6367	77	Trypsin-like serine protease
	SP25/Melanization Protease 1	CG1102	70	Trypsin-like serine protease
	SP145/Jon65Ai	CG10475	58	Chymotrypsin-like serine protease
	SP52	CG8952	56	Elastase-like serine protease
Band G4	SP151	CG18180	459	Elastase-like serine protease
	SP98/Jon65Aiii	CG6483	386	Elastase-like serine protease
	SP143/Jon99Ciii	CG31362	309	Chymotrypsin-like serine protease
	SP88/epsilonTry	CG18681	297	Trypsin-like serine protease
	SP117/Jon65Aiv	CG6467	295	Chymotrypsin-like serine protease
	SP72/Tequila/Graal	CG4821	277	Trypsin-like serine protease
	SP171/Jon66Ci	CG7118	267	Elastase-like serine protease
	SP181/Jon99Fi	CG18030	229	Chymotrypsin-like serine protease
	SP116/yip7/machete	CG6457	210	Chymotrypsin-like serine protease
	SP177/Jon25Biii	CG8871	196	Chymotrypsin-like serine protease
	SP152	CG18179	188	Chymotrypsin-like serine protease
	SP78	CG10472	180	Chymotrypsin-like serine protease
	SP165/Jon66Cii	CG7170	163	Chymotrypsin-like serine protease
	SP83	CG17571	152	Trypsin-like serine protease
	SP119/betaTry	CG18211	143	Trypsin-like serine protease
	SP139/thetaTry	CG12385	141	Trypsin-like serine protease
	SP153/Jon65Aii	CG6580	140	Elastase-like serine protease
	SP177/Jon25Bii	CG8869	132	Chymotrypsin-like serine protease
	SP40	CG4613	59	Trypsin-like serine protease
	SP118	CG34458	53	Chymotrypsin-like serine protease

When a protease was found several times in a band only the best mascot score was reported.

**Table 3 insects-15-00234-t003:** Serine proteases found in silver-stained bands equivalent to *D. melanogaster* Nasrallah hemolymph caseinase bands C1–C3 from Figure 4.

	Name	CG	Mascot Score	Type
Band C1	SP117/Jon65Aiv	CG6467	72	Chymotrypsin-like serine protease
	SP72/Tequila/Graal	CG4821	50	Trypsin-like serine protease
Band C2	SP117/Jon65Aiv	CG6467	446	Chymotrypsin-like serine protease
	SP21	CG3355	137	Trypsin-like serine protease
Band C3	SP72/Tequila/Graal	CG4821	797	Trypsin-like serine protease
	SP143/Jon99Ciii	CG31362	557	Chymotrypsin-like serine protease
	SP145/Jon65Ai	CG10475	346	Chymotrypsin-like serine protease
	SP177/Jon25Bii	CG8869	330	Chymotrypsin-like serine protease
	SP137/Jon25Bi	CG8867	291	Elastase-like serine protease
	c-SP1/grass	CG5896	254	Trypsin-like serine protease
	SP181/Jon99Fi	CG18030	254	Chymotrypsin-like serine protease
	SP116/yip7/machete	CG6457	208	Chymotrypsin-like serine protease
	SP177/Jon25Biii	CG8871	174	Chymotrypsin-like serine protease
	SP33/spirit	CG2056	173	Trypsin-like serine protease
	SP34	CG9372	159	Trypsin-like serine protease
	SP7/Melanization Protease 2	CG3066	151	Trypsin-like serine protease
	SP117/Jon65Aiv	CG6467	132	Chymotrypsin-like serine protease
	SP21	CG3355	122	Trypsin-like serine protease
	SP25/Melanization Protease 1	CG1102	114	Trypsin-like serine protease
	SP153/Jon65Aii	CG6580	111	Elastase-like serine protease
	Hayan	CG6361	100	Trypsin-like serine protease
	SP171/Jon66Ci	CG7118	69	Elastase-like serine protease
	SP91	CG16749	68	Chymotrypsin-like serine protease

When a protease was found several times in a band only the best mascot score was reported.

## Data Availability

The proteomics raw data are deposited in the PRIDE Archive (https://www.ebi.ac.uk/pride/archive/). The project name is: *Drosophila melanogaster* hemolymph gelatinase; project accession number: PXD045527.

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
