# Peer review of "In Drosophila Hemolymph, Serine Proteases Are the Major Gelatinases and Caseinases"

_insects, 2024, doi:10.3390/insects15040234_

Round 1

Reviewer 1 Report

Comments and Suggestions for Authors

Comments on the ms: “In Drosophila hemolymph, serine proteases are the major gelatinases and caseinases” by Gatti et al., for publication in Insects.

In this study the authors used gel zymography to analyze gelatinase and caseinase activities in Drosophila larval hemolymph. They performed mass spectrometry and identified that the major gelatinase and caseinase in Drosophila are serine proteases (SP) but not MMPs. They also analyzed the presence of SPs in the larval hemolymph under divers conditions (bacterial infection, wasp parasitism, .. ) and did not observe differences compared to the control.

This study is interesting, but the conclusions drawn are very limited, and several additional analyses are required to confirm the proposed conclusions. In addition, there is a great deal of variability in the bands (position, intensity) from one gel to another, which makes it difficult to draw conclusions. Finally, it would be very interesting to complete this study by identifying which larval tissues secrete these SPs, and whether changes are observed in response to immune challenges such as bacterial and/or wasp infections.

Comments:

-To help the reader’s comprehension, please indicate on each gel the 6 bands corresponding to gelatinase activity and the 3 bands corresponding to caseinase activity. In many figures, these indications are not given. In Fig. 1b, the positions of the caseinolytic bands are not indicated.

- Why is there such a big difference in the intensity of the bands from one gel to the other? See Fig. 1a and Fig. 1c for Nas: the intensity of the c, d and f bands is very different in these 2 gels! In addition, the band profile for Nas in Fig. 2 is different from that given in Fig. 1a. In order to better trust the data concerning the gelatinase bands identified in other Drosophila species, it is necessary to provide another gel where the Nas profile better matches the pattern given in Fig. 1a.

Fig. S9a and S9b: the pattern of bands is different, why?

-The authors should discuss the wide diversity of gelatinolytic bands between strains of D. melanogaster (Fig. S2). What is the significance of this wide diversity? Why are SP7 and SPE detected differentially in Nas and Canton S (Fig. S5 and Table S3)?

-Fig. 2b and Fig. 5: Please indicate on the figure whether or not there are statistical differences. The measurement of the intensity of the band must not be an absolute value but must correspond to a ratio relative to an internal control in order to avoid any discrepancy due to a difference in loading.

-Table 1: Quantifications are missing.

- What is the correspondence between bands a-f (Fig. 1a) and G1- G4 (Fig. 4b)? This has to be indicated in the text.

-Table 2, why are many proteases present in all the bands? Do they correspond to different isoforms and/or truncated versions? This point needs to be clarified and discussed.

- Fig. S8 the right panel, this gel is not good enough, since in the Nas control the 6 bands are not visible.  A better gel is needed to conclude whether or not there is a difference between the control and the tequila mutant (Iso 1).

-There are discrepancies between published data on the regulation of Tequila after fungal or bacterial infection and the lack of difference observed in this study. This needs to be discussed in order to clarify the limitations (including sensitivity of detection, quantifications,…) of the reported study.

- “Absence of MMP in larval hemolymph” (line 598): to confirm this point, the authors need  to perform  western blots using MMP1 and MMP2 antibodies.

 - An interesting additional piece of information missing from this study concerns the tissues that produce and secrete these proteases. This question can be addressed using reporter lines. It will be very interesting to determine whether SP expression is modified following immune challenges.

Author Response

Response to comments on the ms: “In Drosophila hemolymph, serine proteases are the major gelatinases and caseinases” by Gatti et al., for publication in Insects.

We would like to thank both reviewers for taking the time to read our manuscript and for their very helpful comments and suggestions. We have taken in account most of them and tried to answer their questions in the best possible way.

Reviewer 1

In this study the authors used gel zymography to analyze gelatinase and caseinase activities in Drosophila larval hemolymph. They performed mass spectrometry and identified that the major gelatinase and caseinase in Drosophila are serine proteases (SP) but not MMPs. They also analyzed the presence of SPs in the larval hemolymph under divers conditions (bacterial infection, wasp parasitism, .. ) and did not observe differences compared to the control.

This study is interesting, but the conclusions drawn are very limited, and several additional analyses are required to confirm the proposed conclusions. In addition, there is a great deal of variability in the bands (position, intensity) from one gel to another, which makes it difficult to draw conclusions. Finally, it would be very interesting to complete this study by identifying which larval tissues secrete these SPs, and whether changes are observed in response to immune challenges such as bacterial and/or wasp infections. 

We hope we have answered and clarified on these in the following responses to comments.

Comments:

-To help the reader’s comprehension, please indicate on each gel the 6 bands corresponding to gelatinase activity and the 3 bands corresponding to caseinase activity. In many figures, these indications are not given. In Fig. 1b, the positions of the caseinolytic bands are not indicated. 

The bands a-f were lettered on different gels to make clearer the comparison.

- Why is there such a big difference in the intensity of the bands from one gel to the other? See Fig. 1a and Fig. 1c for Nas: the intensity of the c, d and f bands is very different in these 2 gels! In addition, the band profile for Nas in Fig. 2 is different from that given in Fig. 1a. In order to better trust the data concerning the gelatinase bands identified in other Drosophila species, it is necessary to provide another gel where the Nas profile better matches the pattern given in Fig. 1a.

This is one puzzling problem encounter during our study. We tried to be as much as repetitive as we could, however it was very difficult to get rid of the variations in band intensity between experiments. One reason may be technical such as gel repetition (gelatin changing the gel polymerization) and coloration that can be more or less intense even with the same protocol. Another may be variation between larvae batches, although we tried to work with synchronized middle aged L2, it is difficult to take exactly same age L2 without clear external phenotype, and since hemolymph protease changes may occur in few hours (for ex between late L2 and young L3), this may explain part of the variations.

This is also why we test the hemolymph with and without reducing agent (Fig S3) and observed a peculiar behavior with the D. melanogaster hemolymph. Based on this observation we suggest in the discussion several explanations for these variations, including the possible (and uncontrolled) formation of a clot during the collection and degradation of the collected proteins due to the activation of the many proteases present in hemolymph. It is difficult to block these phenomena although we tried to cool and treat the samples as soon as they were collected. It was not possible to include SDS in the collection media (that will stop or decrease reversibly the protease activity) since this could have broken the larvae tissues and led to high contamination and for obvious reasons, we could not use protease inhibitors.

A second gel for Drosophila species is provided in the supplementary data (Fig S2).

Fig. S9a and S9b: the pattern of bands is different, why?

We change Fig S9 by showing the results with D. melanogaster NasR that was on the same gel than the YR. The Fig S9a showed hemolymph obtained 24h after parasitism so the larvae are almost or at the stage 3 (as shown by the LSPs in the ISy lane) and thus this is different from Fig S9b that showed the mixture of Drosophila L2 hemolymph with Leptopilina venom.

-The authors should discuss the wide diversity of gelatinolytic bands between strains of D. melanogaster (Fig. S2). What is the significance of this wide diversity? Why are SP7 and SPE detected differentially in Nas and Canton S (Fig. S5 and Table S3)?

We added a sentence and a reference on a possible reason of diversity on the gelatinolytic bands between strains/population of D. melanogaster, although this needs certainly more research particularly to understand the significance of the observed differences.

The NasR/CantonS protease differences was due to the bands analyzed in this experiment but overall, the same SPs are retrieved in both strains, we removed this sentence that was confusing and not well explained.

-Fig. 2b and Fig. 5: Please indicate on the figure whether or not there are statistical differences. The measurement of the intensity of the band must not be an absolute value but must correspond to a ratio relative to an internal control in order to avoid any discrepancy due to a difference in loading.

We agree that an absolute ratio will be better, but it is difficult to have an internal marker in this type of experiment. However, since for the inhibition test we loaded the top of the gel with the same sample before cutting the gel slice, we assumed that the proteins are evenly distributed and each slice had the same protein concentration. We agree that this is less evident for the LPS injections. Anyway, we loaded the same amount of hemolymph for each LPS sample, that should allow for a fair comparison. We replace the word measure by estimation.

-Table 1: Quantifications are missing. 

As explained in the math and meth and in the text, the table reports results of a visual estimation based on all bands from the gel slice, and as written in the table footnote, each sign in one cell represents the result from one experiment.

- What is the correspondence between bands a-f (Fig. 1a) and G1- G4 (Fig. 4b)? This has to be indicated in the text. 

This has been added in the text.

-Table 2, why are many proteases present in all the bands? Do they correspond to different isoforms and/or truncated versions? This point needs to be clarified and discussed.

This is part of the discussion (from L250). It is classical in mass spectrometry to find the same protein in different bands, this is due to the nature of the migration of the proteins in an SDS-PAGE. We agree that different form/ isoforms may be present, but this needs a further study to be clarified and is beyond the scope of this manuscript.

- Fig. S8 the right panel, this gel is not good enough, since in the Nas control the 6 bands are not visible.  A better gel is needed to conclude whether or not there is a difference between the control and the tequila mutant (Iso 1). 

We add the band letters and we showed another gel that is pretty similar, and even if some bands may look more intense in Iso1 strain hemolymph the overall band pattern is not clearly different from the wild type line, and we were expected a decrease or an absence of one or more band.

-There are discrepancies between published data on the regulation of Tequila after fungal or bacterial infection and the lack of difference observed in this study. This needs to be discussed in order to clarify the limitations (including sensitivity of detection, quantifications,…) of the reported study.

We agree but the different studies use different approaches. The genes and proteins expression may not be directly correlated, and the two studies did not use the same timing. We did not test for tequila (or other SPs) gene expression after injection, this was not in the scope of the manuscript, but could be develop in the future.

We include in the discussion a part on the limitation of our study with zymogram, particularly when a large quantity of proteases is present in the studied fluid (from L514 and L525).

- “Absence of MMP in larval hemolymph” (line 598): to confirm this point, the authors need  to perform  western blots using MMP1 and MMP2 antibodies.

We remove this from the last sentence. Although we did not find MMPs in hemolymph by mass spectrometry, a western blot could indeed confirm or infirm this, if we assume that western blot is more sensitive than mass spectrometry. However this needs to obtain and test antibodies before, which could be time consuming and may be not necessary for this manuscript. We thus moderate our claim by writing that MMPs are not major drosophila hemolymph gelatinases. (To our knowledge they were not found in other proteomic studies of hemolymph either).

 - An interesting additional piece of information missing from this study concerns the tissues that produce and secrete these proteases. This question can be addressed using reporter lines. It will be very interesting to determine whether SP expression is modified following immune challenges.

We agree but many information are already available in the database on tissues gene expression such as FlyAtlas2 (thanks to the drosophila model), and we used them for the discussion. We also try to test UAS-GAL lines to knock out some of the proteases. We were disappointed since it was impossible to have a larval phenotype with the UAS-GAL lines that allow to select the KO one at stage L2, and the only stable deleted lines for a SP we obtained was Iso 1. We thus think that may be a good continuation for this manuscript but there is a lot of work to do.

Reviewer 2 Report

Comments and Suggestions for Authors

The study has a good conceptualization of research and has a good methodology. However, it is written with lot of details and authors discuss and gave comments in all sections (Introduction, Methodology, Results), some of this comments are needless for the aim and conclusions of the study, they burden the manuscript and making confusion. The whole manuscript needs to be shortened and without repeating the same data in results and discussion section. Please list obtained results in Result section without discussion and comparison with previous studies. In first part of discussion section you repeated a lot of results and gave general statements and conclusions, please rewrote this section without repetition of results, and at the end of the Discussion section, state specific conclusions of your study.

Some minor points:

Line 22: The name of genus should be written in italic

Line 87-96: There is some mistake. You wrote here the discussion and even conclusion about your results. Please write the aim and objectives of the study here, and this section move to the discussion/conclusion part of manuscript.

Line 132: add the explanation of the abbreviation throughout the whole manuscript (eg. IR is explained in line 281, but need to be added on place where mentioned first time)

Line 230: ARN?  Do you mean RNA?

Explain white line-like artefact on Figure 2.

Author Response

Response to comments on the ms: “In Drosophila hemolymph, serine proteases are the major gelatinases and caseinases” by Gatti et al., for publication in Insects.

We would like to thank both reviewers for taking the time to read our manuscript and for their very helpful comments and suggestions. We have taken in account most of them and tried to answer their questions in the best possible way.

Reviewer 2

The study has a good conceptualization of research and has a good methodology. However, it is written with lot of details and authors discuss and gave comments in all sections (Introduction, Methodology, Results), some of this comments are needless for the aim and conclusions of the study, they burden the manuscript and making confusion. The whole manuscript needs to be shortened and without repeating the same data in results and discussion section. Please list obtained results in Result section without discussion and comparison with previous studies. In first part of discussion section you repeated a lot of results and gave general statements and conclusions, please rewrote this section without repetition of results, and at the end of the Discussion section, state specific conclusions of your study.

We tried to follow these suggestions and made efforts to separate results and discussion and remove needless comments. We tried to shorten the introduction and discussion by removing the unnecessary information or results repetition.

Some minor points: 

Line 22: The name of genus should be written in italic

Done, and corrected in the text.

Line 87-96: There is some mistake. You wrote here the discussion and even conclusion about your results. Please write the aim and objectives of the study here, and this section move to the discussion/conclusion part of manuscript.

This was reduced to be not redundant.

Line 132: add the explanation of the abbreviation throughout the whole manuscript (eg. IR is explained in line 281, but need to be added on place where mentioned first time)

IR is now explained line 118, the first time it is used.

Line 230: ARN?  Do you mean RNA? 

 Corrected, thanks.

Explain white line-like artefact on Figure 2. 

Done

Round 2

Reviewer 1 Report

Comments and Suggestions for Authors

The revised version if ready for publication.  

Author Response

Thank you.

Reviewer 2 Report

Comments and Suggestions for Authors

In the attached document.

Author Response

Thank you for your demand. We have uploaded a file "Gatti et al, corrected with mark.doc" with marks as a non publishable material.

Best regards 

Round 3

Reviewer 2 Report

Comments and Suggestions for Authors

The authors improved manuscript a lot. They accepted all suggestions. However, appropriate conclusion section still missing. The written conclusions are general statements without clear underlining of your concrete conclusions. Please write concrete conclusions which are drown from your results.

Author Response

Dear reviewer

As suggested we made a conclusion based on the data reported and on the way they pave for future research. We hope it is what the reviewer expected.

Best regards